# Bringing Saccades and Fixations into Self-supervised Video Representation Learning

## Abstract

In this paper, we propose a self-supervised video representation learning (video SSL) method by taking inspiration from cognitive science and neuroscience on human visual perception. Different from previous methods that focus on the inherent properties of videos, we argue that humans learn to perceive the world through the self-awareness of the semantic changes or consistency in the input stimuli in the absence of labels, accompanied by representation reorganization during the post-learning rest periods. To this end, we first exploit the presence of saccades as an indicator of semantic changes in a contrastive learning framework, mimicking the self-awareness in human representation learning. The saccades are generated artificially without eye-tracking data. Second, we model the semantic consistency in eye fixation by minimizing the prediction error between the predicted and the true state of another time point. Finally, we incorporate prototypical contrastive learning to reorganize the learned representations to enhance the associations among perceptually similar ones. Compared to previous video SSL solutions, our method can capture finer-grained semantics from video instances, and the associations among similar ones are further strengthened. Experiments show that the proposed bio-inspired video SSL method significantly improves the Top-1 video retrieval accuracy on UCF101 and achieves superior performance on downstream tasks such as action recognition under comparable settings.

## 1 Introduction

Learning without labels is the most common way for humans to get to know the world (DiCarlo et al., 2012), and it has also been widely studied in machine learning for developing intelligent agents. In particular, many researchers focus on self-supervised learning (SSL) from dynamic visual input data, *i.e.*, videos (Hurri & Hyvärinen, 2003; Mobahi et al., 2009; Srivastava et al., 2015), which comes closest to the natural data perceived by humans. Recently, deep learning based video SSL methods have also shown superior performance over traditional non-deep learning methods (Wang et al., 2021a; Duan et al., 2022). However, there is still a large room for improvement considering the gap between the unsupervised learning abilities of deep learning models and humans.

One notable difference between deep video SSL methods and human unsupervised learning is that the former typically learn discriminative representations by considering the inherent data properties, such as the clip order (Misra et al., 2016), the spatiotemporal coherence (Vondrick et al., 2018), the transformations exerted (Jenni et al., 2020), *etc.*, and propose various pretext tasks accordingly. While for humans, the self-awareness of the **semantic change or consistency** in the input stimuli is essential for learning without labels (Melcher & Colby, 2008). Besides, the encoded representations in the brain are not left unchanged but kept being **reorganized** to yield a representation structure with strengthened associations among perceptually similar representations (Diekelmann & Born, 2010). Fig. 1 shows an overall comparison. This discrepancy inspired us to propose a new video SSL method by taking inspiration from cognitive science and neuroscience on human visual perception. Recently, Illing et al. (2021) proposed a bio-inspired unsupervised learning rule that treats the presence of saccades as a global synaptic modulator. However, it is less powerful due to the inherent difficulty in optimizing deep networks with layer-wise optimization.

Human visual perception is mainly accomplished by alternating *saccade* and *fixation* when the heads are relatively still. The former is the rapid foveal motion from one target of interest to another,

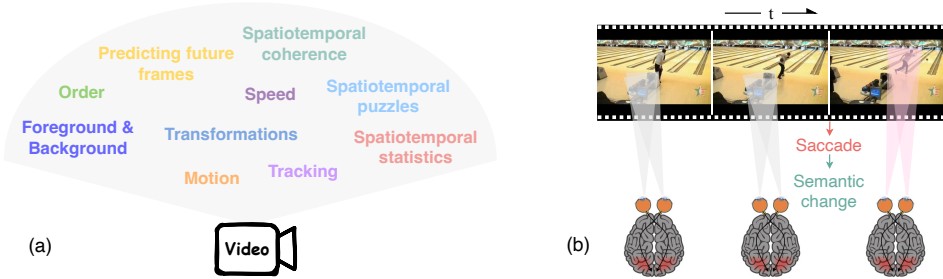

Figure 1: **Overall comparison. (a) Previous video SSL methods** design pretext tasks based on the inherent properties of videos, while **(b) our method** explores the presence of saccades as an indicator of semantic change to mimic the role of self-awareness in human perception. Since it is relatively expensive to collect real gaze data, we construct artificial saccades for training (§3.1).

while the latter is the period where the eye is kept aligned with one target for processing visual details. To capture the **semantic change** in the video, we propose to exploit the presence of saccades as an indicator of the semantic change and propose a semantic-change-aware contrastive learning framework. This is inspired by the fact that the human would perform a saccade when a semantic change occurs in the fixation area. Specifically, the positive pairs are formed by features of the same fixation location in a video, and the negative pairs are formed by features of different fixation locations in the same video or features from different videos. Compared to previous contrastive-based video SSL methods, our method captures finer-grained semantics within the same video. Note that we manually construct saccades by exerting different fixation masks on the input without using real gaze data, making our method a general one for any video data without extra supervision.

We further encourage the **semantic consistency** within a fixation duration by minimizing the prediction error (PE) when using the current state to predict that of another time point in the same fixation duration. In this way, PE can serve as an extra supervision signal to avoid semantic discrepancy during semantic-change-aware contrastive learning. This is also biorational, as PE is known as an important modulator in perception, attention, and motivation control (Den Ouden et al., 2012).

To enhance the association among the previously learned finer-grained semantics, inspired by the **reorganization** in human representation learning (Diekelmann & Born, 2010), we incorporate prototypical contrastive learning (Li et al., 2020) to gradually redistribute the representations. The learned representations are pulled towards their corresponding prototypes and pushed away from other prototypes. Such post-learning reorganization facilitates grouping unseen input stimuli into meaningful categories based on similarity, which leads to improved Top-1 retrieval accuracy compared with previous contrastive-based video SSL methods.

In summary, we propose a video SSL framework by taking inspiration from cognitive science and neuroscience on human visual perception. We first exploit the presence of saccades as an indicator of semantic change in a contrastive learning framework for modeling the role of self-awareness in human representation learning. Then, we model the semantic consistency in the input by minimizing PE between a predicted and the true states of different time points during a fixation. Third, we incorporate prototypical contrastive learning to reorganize the learned representations such that the associations among perceptually similar ones would be strengthened after redistribution. Experiments show that the proposed bio-inspired video SSL method significantly improves the Top-1 video retrieval accuracy on UCF101, and achieves superior performance on downstream tasks such as action recognition. The code and the pre-trained models will be released.

## 2 RELATED WORK

Self-supervised learning has been studied with various data formats, including image (Wu et al., 2018; Grill et al., 2020; He et al., 2020; Li et al., 2020), video (Xu et al., 2019; Benaim et al., 2020; Qian et al., 2021a; Duan et al., 2022), and multi-modal data (Alayrac et al., 2020; Patrick et al., 2021). In this section, we focus on self-supervised representation learning on videos.

**Non-contrastive video SSL methods.** Previous video SSL methods mainly learn discriminative representations by designing various *pretext tasks* based on the analysis of the inherent spatiotemporal properties of video data. The pretext tasks include figuring out the correct order of

the clips (Misra et al., 2016; Fernando et al., 2017; Lee et al., 2017; Wei et al., 2018; Xu et al., 2019), tracking contents across adjacent frames (Wang & Gupta, 2015; Vondrick et al., 2016; Pathak et al., 2017; Vondrick et al., 2018; Wang et al., 2019b), studying foreground and background robustness (Luo et al., 2017; Wang et al., 2021b;c; Ding et al., 2022), predicting future frames (Vondrick et al., 2016; Luo et al., 2017; Villegas et al., 2017; Han et al., 2020a; Behrmann et al., 2021), solving spatiotemporal puzzles (Kim et al., 2019) or video cloze (Luo et al., 2020), learning the spatiotemporal statistics of the videos (Wang et al., 2019a; 2021a), recognizing transformations exerted on the video (Jenni et al., 2020; Duan et al., 2022) , or determining whether a video is played at the intrinsic speeds (Benaim et al., 2020; Wang et al., 2020; Yao et al., 2020; Chen et al., 2021).

**Contrastive video SSL methods.** Another line of research adapts spatiotemporal properties into the *contrastive learning* framework (Hadsell et al., 2006) by constructing the positive and negative pairs based on various spatiotemporal cues. More specifically, some methods extend the instance discrimination methods in image SSL (Wu et al., 2018), and directly use clips randomly sampled from the same video as a positive pair (Han et al., 2020b; Lin et al., 2021; Pan et al., 2021; Qian et al., 2021b; Yao et al., 2021). Some methods consider the spatiotemporal consistency of video and treat the predicted and the ground-truth features at the same spatiotemporal location as a positive pair (Han et al., 2019; 2020a). Some methods relate video understanding to pace reasoning ability, and construct positive pairs by sampling clips with different sampling rates from the same video or sampling clips with the same pace from different videos (Huang et al., 2021). Some methods construct the positive pair from both frame-level and video-level representations (Kong et al., 2020; Kuang et al., 2021). Some others construct positive pairs through spatiotemporal data augmentations (Qian et al., 2021b; Sun et al., 2021), or exploit motion information for data augmentation (Dwibedi et al., 2019; Li et al., 2021; Wang et al., 2021b). Our method also belongs to this line of research. However, we construct positive and negative pairs based on the semantic change indicated by saccades, resulting in a finer-grained distinction of the semantics from the same video instance.

**Comparison with CLAPP.** Both the CLAPP model (Illing et al., 2021) and our work exploit the self-awareness of saccades for self-supervised representation learning, but we differ significantly in several aspects. First, CLAPP aims to propose a local learning rule for building deep representations without back-propagation, where each layer is trained independently to predict whether a saccade happens. While ours utilizes the presence of saccades to construct the positive and negative pairs in a contrastive learning framework for end-to-end learning. Second, for each layer, CLAPP restricts the predictions to be similar to its responses to future inputs and as different as possible from its responses to fake inputs. However, our method minimizes the discrepancies between the predictions and the future responses in the absence of a saccade for semantic consistency modeling. Third, besides constructing a saccade by switching the network inputs from one video to another as in CLAPP, we also consider inter-video saccades, which are realized by intentionally changing the fixation area on the same video to capture finer-grained semantics. As shown in §4, we significantly improve over CLAPP for the video recognition task on UCF101.

## 3 METHOD

As shown in Fig. 2, our method consists of three parts. First, we explore saccades as the indicator of semantic change in a contrastive learning process (§3.1), where the negative pairs consist of features before and after a saccade occurs. Second, we model the semantic consistency during fixation by minimizing the prediction error (PE) between the predicted and the true features of different time points (§3.2). Third, we perform a post-learning reorganization to strengthen the associations among perceptually similar representations (§3.3).

### 3.1 CAPTURING SEMANTIC CHANGE VIA CONTRASTIVE LEARNING

**Preparing saccades.** The ground-truth gaze data are generally collected using eye-tracking devices, which requires a lot of manual effort, and usually has personal heterogeneity considering the exact fixation locations. To mitigate these problems, we propose to construct artificial saccades, and only consider a representative set of coarse fixation locations. We simulate the receptive field of the fovea by exerting fixation masks $\{\mathbf{m}_i\}_{i=1}^{N_p}$ on the input stimuli. The artificial saccades are constructed by intentionally alternating the fixation masks. To balance the performance and efficiency, we set $N_p = 5$. The spatial size of fixation area is 1/4 of the whole input as shown in Fig. 3 (a), and the temporal length is the same as the clip length. More details are presented in §A.1.

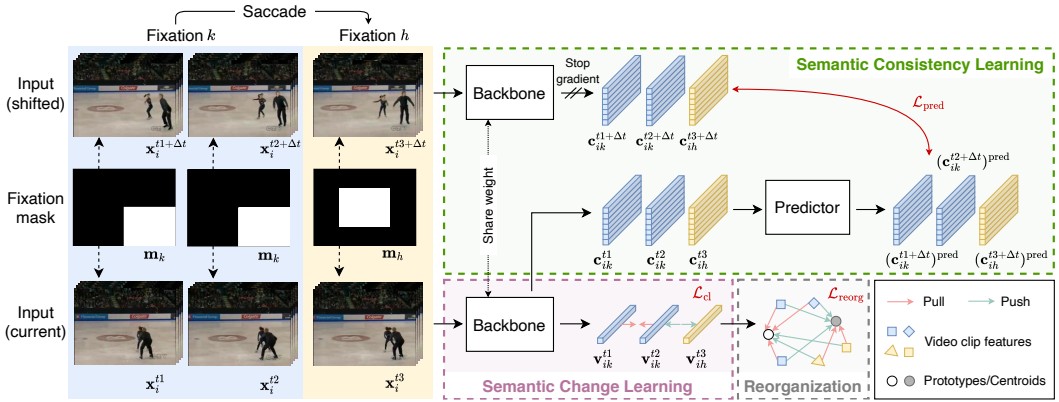

Figure 2: **Framework of the proposed bio-inspired video SSL method.** Given a video $\mathbf{x}_i$, the positive and negative pairs for semantic-change-aware contrastive learning can be constructed by exerting the same or different fixation masks $\mathbf{m}$ on different clips of the video (§3.1), respectively. Further, the semantic consistency is modeled by minimizing the prediction error between the predicted state $(\mathbf{c}_{ik}^{t*+\Delta t})^{\text{pred}}$ and the true state of another time point $\mathbf{c}_{ik}^{t*+\Delta t}$ (§3.2). Finally, the learned representations are reorganized through a prototypical contrastive learning process to strengthen the association among perceptually similar ones (§3.3).

**Semantic-change-aware contrastive learning.** To capture the semantic change in the presence of a saccade, we try to make the representations before and after a saccade distinct from each other. To this end, we treat the representations before and after a saccade as negative pairs, and otherwise positive pairs. The latent feature space is thus optimized by minimizing the contrastive loss.

Formally, given a training set $\mathbf{X} = \{\mathbf{x}_1, \mathbf{x}_2, \ldots, \mathbf{x}_N\}$ of $N$ videos without category labels, we aim to learn an embedding function $f_\theta$ to map $\mathbf{X}$ to features $\mathbf{V} = \{\mathbf{v}_1, \mathbf{v}_2, \ldots, \mathbf{v}_N\}$, where $\mathbf{v}_i = f_\theta(\mathbf{x}_i)$ and $\mathbf{v}_i \in \mathbb{R}^D$ is expected to capture the semantics of $\mathbf{x}_i$. The semantic-change-aware contrastive loss is inspired by InfoNCE (Oord et al., 2018; He et al., 2020), and is calculated as follows:

$$\mathcal{L}_{\text{cl}} = \frac{1}{N} \sum_{i=1}^{N} -\log \frac{\exp(\mathbf{v}_{ik} \cdot \mathbf{v}'_{ik}/\tau)}{\exp(\mathbf{v}_{ik} \cdot \mathbf{v}'_{ik}/\tau) + \sum_{j \in \mathcal{I}_-} \exp(\mathbf{v}_{ik} \cdot \mathbf{v}'_{jh}/\tau))}, \quad |\mathcal{I}_-| = N_{\text{neg}}. \tag{1}$$

Here, $\mathbf{v}'_{ik} = f_\theta(\mathbf{x}'_i \odot \mathbf{m}_k)$ is a positive sample for $\mathbf{v}_{ik} = f_\theta(\mathbf{x}_i \odot \mathbf{m}_k)$ ($\odot$ is Hadamard product), $\mathbf{x}'_i$ is obtained by applying commonly-used data augmentations on $\mathbf{x}_i$, $\mathbf{v}'_{jh} \neq \mathbf{v}'_{ik}$ is a negative sample, $\mathcal{I}_-$ is the set of indices for $N_{\text{neg}}$ selected negative samples, and $k, h \in \{1, 2, \ldots, N_p\}$ are indices of the mask $\mathbf{m}$. Note that a sample is considered positive if and only if $j = i$ and $h = k$, i.e., they are from the same fixation region of the same video. By minimizing Eq. (1), the embedding function $f_\theta$ is trained to distinguish between finer-grained semantics within a video. Perceptually similar finer-grained semantics will be further associated together through a reorganization process.

**Memory bank.** As previously revealed in (Oord et al., 2018; Wu et al., 2018), a large number of negative pairs is essential for training InfoNCE loss, which is typically restricted by the batch size. To alleviate this issue, we follow (Wu et al., 2018) and maintain a memory bank $\overline{\mathbf{V}} = \{\overline{\mathbf{v}}_i\}_{i=1}^{N*N_p}$ for different fixation locations of all the videos in the training dataset. Here, $N$ is the number of training videos, and $N_p$ is the number of masks. Similar to (Wu et al., 2018), we initialize $\overline{\mathbf{V}}$ with random $D$-dimensional unit vectors and update the slot $\overline{\mathbf{v}}_i$ with the latest feature $\mathbf{v}_i$ as follows:

$$\overline{\mathbf{v}}_i \leftarrow (1-m)\overline{\mathbf{v}}_i + m\mathbf{v}_i, \tag{2}$$

where $m \in [0, 1]$ is a momentum value. With $\overline{\mathbf{V}}$, we can rewrite the contrastive learning procedure and Eq. (1) by replacing the negative samples $\mathbf{v}'$ by their memory bank representations $\overline{\mathbf{v}}$.

### 3.2 Modeling semantic consistency via minimizing PE

To encourage the semantic consistency between the states of two time points within the a fixation, given a video $\mathbf{x}_i$, we minimize the PE when using $\mathbf{c}_i^t$ to predict $\mathbf{c}_i^{t+\Delta t}$, where $\mathbf{c}_i^* \in \mathbb{R}^{C_l \times (T_l H_l W_l)}$

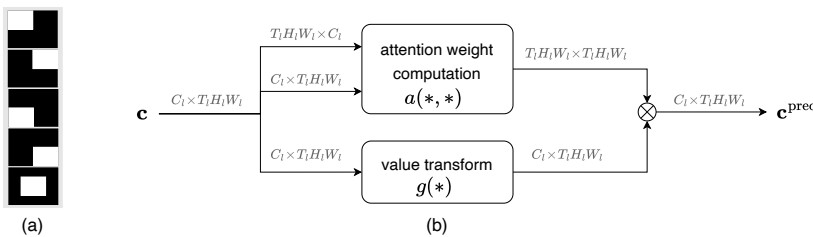

Figure 3: Method details. **(a) Exemplar fixation masks** for $N_p = 5$, which cover the major region in the visual field. See §3.1) for more details. **(b) Prediction module** introduced in §3.2), which is designed as a variant of the self-attention module to capture the spatiotemporal correlations.

is reshaped from the $l$-th level feature from the backbone $f_\theta$ whose size is $C_l \times T_l \times H_l \times W_l$, and $C_l, T_l, H_l, W_l$ are the feature channel number, the temporal resolution, the height, and the width, respectively. Note that $t$ is the time point of a clip from $\mathbf{x}_i$, and $\Delta t$ can be either positive or negative.

The predictor module $p_\theta$ is designed as a variant of the self-attention module (Bahdanau et al., 2015; Xie et al., 2021) to capture the spatiotemporal correlations. It first obtains the spatiotemporal weights based on the element-wise cosine similarity, and then calculates each predicted element as the weighted sum of the transformed version of all other spatiotemporal elements. An illustration is shown in Fig. 3 (b). The $u$-th element of the predicted feature $(\mathbf{c}_i^{t+\Delta t})^{\text{pred}}$ is calculated as:

$$(\mathbf{c}_{iu}^{t+\Delta t})^{\text{pred}} = \sum_{v=1}^{T_l H_l W_l} a(\mathbf{c}_{iu}^t, \mathbf{c}_{iv}^t) \cdot g(\mathbf{c}_{iv}^t), \tag{3}$$

where $a(\cdot, \cdot)$ calculates the attention weights as:

$$a(\mathbf{c}_{iu}^t, \mathbf{c}_{iv}^t) = \text{ReLU}(\cos(\mathbf{c}_{iu}^t, \mathbf{c}_{iv}^t)), \tag{4}$$

and the transform function $g(\cdot)$ is a linear layer that maps a $C_l$-dim input to a $C_l$-dim output. The semantic consistency is optimized by minimizing the prediction loss defined as follows:

$$\mathcal{L}_{\text{pred}} = \frac{1}{N} \sum_{i=1}^N \frac{1}{T_l H_l W_l} \sum_{u=1}^{T_l H_l W_l} |\mathbf{c}_{iu}^{t+\Delta t} - (\mathbf{c}_{iu}^{t+\Delta t})^{\text{pred}}|. \tag{5}$$

### 3.3 REORGANIZING VIA PROTOTYPICAL CONTRASTIVE LEARNING

Inspired by the gradual redistribution and reorganization of memory representations during the post-learning rest periods (Diekelmann & Born, 2010), after training the whole framework with $\mathcal{L}_{\text{cl}}$ (Eq. (1)) and $\mathcal{L}_{\text{pred}}$ (Eq. (5)) to converge, we further perform reorganization through prototypical contrastive learning (Li et al., 2020) to strengthen the associations among similar representations.

Specifically, we first cluster the representations $\{\mathbf{v}_{ik}\}$ for $R$ times to obtain $R$ distinct clustering results $\{\mathbf{G}^{(r)}\}_{r=1}^R$, where $\mathbf{G}^{(r)}$ contains $Q^{(r)}$ clusters. Then, we randomly pick $Q = \min\{Q^{(r)}, N_{\text{neg}}\}$ clusters from each $\mathbf{G}^{(r)}$ to form $\mathbf{G}^{(r)'} = \{G_q^{(r)}\}_{q=1}^Q$, where the centroid of $G_q^{(r)}$ is $\mathbf{o}_q^{(r)}$. The reorganization loss is calculated as follows:

$$\mathcal{L}_{\text{reorg}} = \frac{1}{N} \sum_{i=1}^N \frac{1}{N_p} \sum_{k=1}^{N_p} \frac{1}{R} \sum_{r=1}^R -\log \frac{\exp(\mathbf{v}_{ik} \cdot \mathbf{o}_s^{(r)}/\phi_s^{(r)})}{\sum_{j=0}^Q \exp(\mathbf{v}_{ik} \cdot \mathbf{o}_j^{(r)}/\phi_j^{(r)}))}. \tag{6}$$

Here, $G_s^{(r)}$ is the cluster to which $\mathbf{v}_{ik}$ is assigned, $\mathbf{o}_s^{(r)}$ is the centroid of $G_s^{(r)}$, and $\phi_*^{(r)}$ denotes the concentration estimation of the cluster $G_*^{(r)}$ as in (Li et al., 2020). In this way, the associations among previously learned finer-grained semantics are further strengthened, which can facilitate similarity-based categorization for unknown input stimuli.

### 3.4 LOSS FUNCTION

The overall loss function is a combination of the three loss terms introduced above:

$$\mathcal{L} = \mathcal{L}_{\text{cl}} + \mathcal{L}_{\text{pred}} + \mathcal{L}_{\text{reorg}}, \tag{7}$$

where the third term only takes effect at the late stage of the training process.

Table 1: **Video retrieval results on UCF101 and HMDB51.** All the models are pre-trained on UCF101. Larger values are better. See §4.2 for details.

| Method | Backbone | UCF101 | | | | HMDB51 | | | |
|---|---|---|---|---|---|---|---|---|---|
| | | Top-1 | Top-5 | Top-10 | Top-20 | Top-1 | Top-5 | Top-10 | Top-20 |
| **Non-contrastive video SSL methods** | | | | | | | | | |
| VCOP | R(2+1)D | 10.7 | 25.9 | 35.4 | 47.3 | 5.7 | 19.5 | 30.7 | 45.8 |
| VCP | R(2+1)D | 19.9 | 33.7 | 42.0 | 50.5 | 6.7 | 21.3 | 32.7 | 49.2 |
| PRP | R(2+1)D | 20.3 | 34.0 | 41.9 | 51.7 | 8.2 | 25.3 | 36.2 | 51.0 |
| Pace | R(2+1)D | 25.6 | 42.7 | 51.3 | 61.3 | 12.9 | 31.6 | 43.2 | 58.0 |
| STS | R(2+1)D | 38.1 | 58.9 | 68.1 | 77.0 | 16.4 | 36.9 | 50.5 | 65.4 |
| VCOP | R3D-18 | 14.1 | 30.3 | 40.0 | 51.1 | 7.6 | 22.9 | 34.4 | 48.8 |
| VCP | R3D-18 | 18.6 | 33.6 | 42.5 | 53.3 | 7.6 | 24.4 | 36.3 | 53.6 |
| PRP | R3D-18 | 22.8 | 38.5 | 46.7 | 55.2 | 8.2 | 25.8 | 38.5 | 53.3 |
| Pace | R3D-18 | 23.8 | 38.1 | 46.4 | 56.6 | 9.6 | 26.9 | 41.1 | 56.1 |
| STS | R3D-18 | 38.3 | _59.9_ | 68.9 | _77.2_ | 18.0 | 37.2 | 50.7 | 64.8 |
| TransRank | R3D-18 | 46.5 | **63.7** | **72.8** | - | **19.4** | **45.4** | **59.1** | - |
| **Contrastive video SSL methods** | | | | | | | | | |
| MemDPC | R18 | 20.2 | 40.4 | 52.4 | 64.7 | 7.7 | 25.7 | 40.6 | 57.5 |
| MLRep | R3D-18 | 39.6 | 57.6 | _69.2_ | **78.0** | 18.8 | 39.2 | 51.0 | 63.7 |
| **Ours** | R3D-18 | **48.5** | 58.6 | 65.3 | 72.1 | 17.6 | 35.7 | 51.4 | 65.5 |
| | R(2+1)D | _47.6_ | 58.8 | 66.2 | 74.6 | _19.0_ | _39.7_ | _54.2_ | **82.8** |

## 4 EXPERIMENTS

### 4.1 IMPLEMENTATION DETAILS

**Datasets.** We conduct experiments on two representative video datasets, namely UCF101 (Soomro et al., 2012) and HMDB51 (Kuehne et al., 2011). UCF101 consists of 13K videos of 101 action classes. HMDB51 contains 7K videos from 51 action classes. Both UCF101 and HMDB51 have three official train/test splits. We pre-trained on the first train split of UCF101 and used the first train/test split of UCF101 and HMDB-51 for evaluation following (Wang et al., 2020; Qian et al., 2021a). For the ablation study, we use the first train/test split of UCF101.

**Network architectures.** We experiment on two backbones that are commonly used in previous video SSL methods, namely R3D-18 (Hara et al., 2018) and R(2+1)D-18 (Tran et al., 2018). Given $N_p$ randomly sampled 16-frame clips of resolution $112 \times 112$ from a video, the backbone outputs $N_p$ $D$-dim feature vectors, where $D = 512$, and $N_p = 5$ is the number of types of fixation masks. We sample $N_p$ clips for a video and ensure that all the $N_p$ memory slots for the video can be updated.

**Pre-training.** The batch size for R3D is 16, and the batch size for R(2+1)D is 14. The two backbones are trained for 300 epochs on the UCF101 training set using SGD with a momentum of 0.9 and weight decay of $10^{-4}$. The initial learning rate is 0.1 and is decayed by 5 at epoch 90, 180, and 240. The number of negative samples $N_{neg} = 1024$. For PE minimization, we set $l = 5$, *i.e.*, the state $\mathbf{c}_i^*$ is from the 5-th level of the backbone network, which shows the best balance between accuracy and efficiency. For reorganization, we use an unsupervised clustering algorithm Faiss (Johnson et al., 2019), and set $R = 3$, $Q^{(r)} = 1500$ for $r = 1, \ldots, R$, and $Q = 1024$ based on ablation study. We train for another 60 epochs after incorporating $\mathcal{L}_{reorg}$ using SGD with a learning rate of $8^{-4}$.

**Action recognition.** We initialize the backbone using the model parameters obtained in the pre-training stage except for the last linear layer. We consider two settings: i) *linear probe*, where only the last linear layer is trained with cross-entropy loss, and ii) *finetune*, where the entire network is finetuned with cross-entropy loss. For *linear probe*, when training on UCF101, we use SGD optimizer and trained for 200 epochs with an initial learning rate of 0.1, which is further decayed by 10 at epoch 60, 120 and 180. For HMDB51, we use Adam with a learning rate of 0.001 and trained to converge. We use batch size 32 for both R3D and R(2+1)D backbones with input resolution $112 \times 112$. For *finetune*, we use an SGD optimizer and trained for 200 epochs with a large initial learning rate of 0.1 following (Duan et al., 2022), which is further decayed by 10 at epoch 60, 120, and 180. The batch size is 32 for both backbones. During training, we apply the same data augmentation as in (Han et al., 2020b). For evaluation, we uniformly sample 10 clips from one testing video, perform the center crop, resize them to $112 \times 112$, and average the predicted probabilities as the final prediction, following (Wang et al., 2020; Qian et al., 2021a).

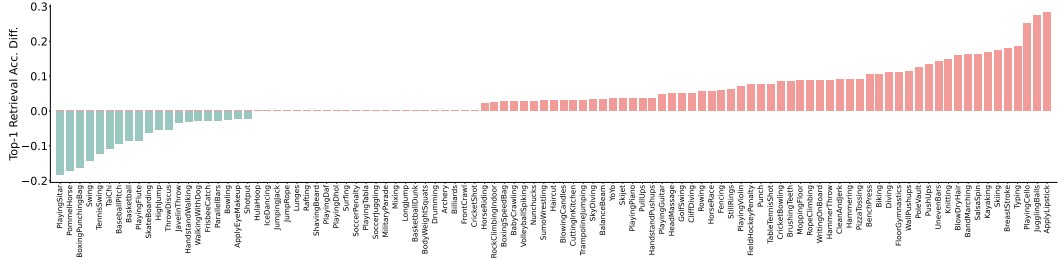

Figure 4: **Top-1 retrieval accuracy comparison before and after reorganization on UCF101.** The reorganization process can improve the Top-1 retrieval performance by 57.4% out of all the classes, leading to comparative performance of 22.8% out of all. See §4 for more discussions.

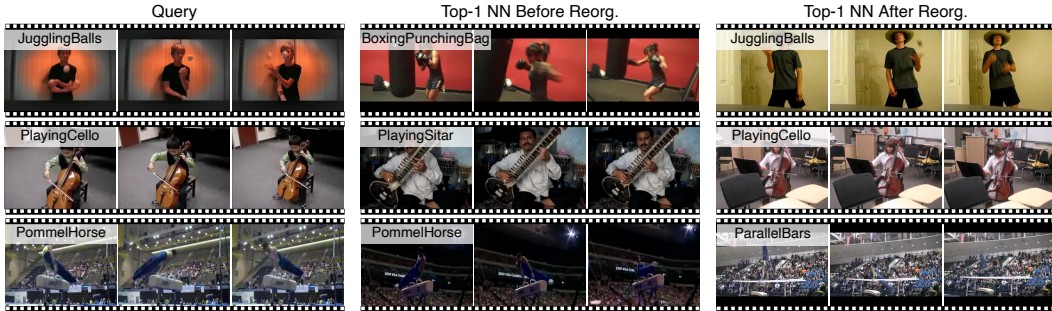

Figure 5: **Qualitative results for video retrieval before and after reorganization.** The first two examples are improved by reorganization, and the third example is a failure case, where the retrieval result after the reorganization is more similar in appearance while different in semantics. See §4.

## 4.2 COMPARISON WITH THE STATE-OF-THE-ART

To evaluate the representations learned in the self-supervised pre-training stage, we perform *video retrieval* and *action recognition* and compare them with other state-of-the-art methods.

**Video retrieval.** We query the $k$-nearest neighbors of the testing set video clips from the training set. The pre-trained backbone model (R3D or R(2+1)D) is directly used as a feature extractor without further finetuning. For each query video, we obtain one 512-d feature vector by extracting and averaging the features of 10 uniformly sampled clips. We use cosine similarity to measure the distance of the features for determining the $k$-nearest neighbors. A correct retrieval is counted when the $k$ nearest neighbors contain at least one video of the same class with the query video. We report Top-$k$ retrieval accuracy in Table 1, where $k = 1, 5, 10, 20$, and compare it with other video SSL methods pre-trained on the RGB modality of UCF101, including VCOP (Xu et al., 2019), VCP (Luo et al., 2020), PRP (Yao et al., 2020), Pace (Wang et al., 2020), STS (Wang et al., 2021a), and TransRank (Duan et al., 2022). As can be observed, our method achieves the best Top-1 retrieval accuracy on UCF101, and comparable Top-1 performance on HMDB51. Though our method is slightly inferior to other Top-$k$ values, we argue that the Top-1 metric matters the most in security-demanded real-world applications such as autonomous driving. Primates including humans are also more good at giving out one promising guess than several candidates (Freedman et al., 2001).

Compared to previous methods, our method captures finer-grained semantics, which are reorganized to yield a better representation structure. To see the change of the learned finer-grained semantics before and after reorganization, we make statistics on the Top-1 retrieval accuracy before and after reorganization on UCF101 regarding the 101 categories. As shown in Fig. 4, the post-learning reorganization improves the Top-1 retrieval accuracy for 57.4% of the 101 action categories. For 22.8% categories, the Top-1 retrieval accuracy remains the same. This shows that the redistributed finer-grained semantics can facilitate similarity based categorization.

We further visualize video retrieval examples that are corrected by reorganization or conversely in Fig. 5. The first two examples are rectified by reorganization. For the `JugglingBalls` query clip, it retrieves a `BoxPunchingBag` clip with similar foreground and background color, *i.e.*, a person

Table 2: **Action recognition results on UCF101 and HMDB51.** The models are pre-trained on UCF101 with RGB only. $^*$ means learning without back-propagation. See §4.2 for details.

| Method | Backbone | Input Size | Frozen | UCF101 | HMDB51 |
|---|---|---|---|---|---|
| CLAPP$^*$ (Illing et al., 2021) | VGG-5 | - | ✓ | 41.6 | - |
| Object Patch (Hadsell et al., 2006) | AlexNet | - | ✓ | 42.7 | 15.6 |
| Seq Ver. (Misra et al., 2016) | CaffeNet | - | ✓ | 50.9 | 19.8 |
| OPN (Lee et al., 2017) | CaffeNet | $1 \times 80^2$ | ✓ | 56.3 | 22.1 |
| Geometry (Gan et al., 2018) | FlowNet | - | ✓ | 54.1 | 22.6 |
| Bi-Geometry (Gan et al., 2018) | FlowNet | - | ✓ | 55.1 | 23.3 |
| **Ours** | R3D-18 | $16 \times 112^2$ | ✓ | **59.2** | **32.0** |
| VCOP (Xu et al., 2019) | R3D-18 | $16 \times 112^2$ | ✗ | 64.9 | 29.5 |
| PRP (Yao et al., 2020) | R3D-18 | $16 \times 112^2$ | ✗ | 66.5 | 29.7 |
| STS (Wang et al., 2021a) | R3D-18 | $16 \times 112^2$ | ✗ | 70.4 | 34.9 |
| VCP (Luo et al., 2020) | R3D-18 | $16 \times 112^2$ | ✗ | 66.0 | 31.5 |
| Pace (Wang et al., 2020) | R3D-18 | $16 \times 112^2$ | ✗ | 77.1 | 36.6 |
| MLRep (Qian et al., 2021a) | R3D-18 | $16 \times 112^2$ | ✗ | 76.2 | 41.1 |
| **Ours** | R3D-18 | $16 \times 112^2$ | ✗ | **76.6** | **43.1** |
| VCOP (Xu et al., 2019) | R(2+1)D | $16 \times 112^2$ | ✗ | 72.4 | 30.9 |
| PRP (Yao et al., 2020) | R(2+1)D | $16 \times 112^2$ | ✗ | 72.1 | 35.0 |
| STS Wang et al. (2021a) | R(2+1)D | $16 \times 112^2$ | ✗ | 77.8 | 40.7 |
| VCP (Luo et al., 2020) | R(2+1)D | $16 \times 112^2$ | ✗ | 66.3 | 32.2 |
| Pace (Wang et al., 2020) | R(2+1)D | $16 \times 112^2$ | ✗ | 75.9 | 35.9 |
| **Ours** | R(2+1)D | $16 \times 112^2$ | ✗ | **78.1** | **44.8** |

in black and red background. For the `PlayingCello` query clip, it retrieves a `PlayingSitar` clip that resembles in both appearance and motion. By representation reorganization, such ambiguousness are removed, and clips with the correct action labels are found. However, we also notice that, for clips with complicated backgrounds, such as `PommeiHorse` in the third row of Fig. 5, reorganization tends to make it easier to be mixed up with clips having similar appearance.

**Action recognition.** We report the Top-1 action recognition accuracy of linear probe (`Frozen` ✓) and finetune (`Frozen` ✗) in Table 2. For a fair comparison, we exclude methods based on much deeper backbones, with larger input resolution, using multi-model data, or pre-trained on much larger video datasets such as Kinetics (Carreira & Zisserman, 2017). For linear probe experiments, our method outperforms all previous video SSL methods pre-trained on UCF101, especially for CLAPP (Illing et al., 2021), a bio-inspired unsupervised representation learning method without back-propagation. This clearly demonstrates that our method is a competitive practice of exploring cognitive inspirations in deep self-supervised representation learning. For the finetune setting, our method achieves the highest Top-1 accuracy compared with other video SSL methods pre-trained on the RGB modality of UCF101, showing a good generalization ability of the learned representations.

### 4.3 ABLATION STUDY

In this section, we assess the effectiveness of the framework design regarding three components: the semantic-change-aware contrastive learning that utilizes saccade as an indicator (§3.1), the semantic consistency learning by minimizing the prediction error (PE)(§ 3.2), and the reorganization via prototypical contrastive learning (§3.3). We report Top-1 video retrieval accuracy on UCF101 to evaluate the learned video representations without further finetuning.

**Overall framework design.** As shown in Table 3, although increasing the size of the memory bank benefits the retrieval performance, major improvements are brought by incorporating the three components. Specifically, incorporating saccades for constructing negative pairs achieves an absolute improvement of 1.7 point, and further including PE minimization or post-learning reorganization leads to an absolute improvement of 0.8 and 2.4, respectively. The full model is 4.3 point higher than the baseline with a memory bank of the same size. The results clearly demonstrate the effectiveness of each framework design.

**Alternatives of artificial saccades.** Besides constructing artificial saccades for training, we tried to incorporate real gaze data in our scheme by resorting to current video saliency prediction datasets such as DHF1K (Wang et al., 2018), Hollywood-2, and UCF sports (Mathe & Sminchisescu, 2015). However, those datasets typically contain no more than 1.5K training videos, which are much

Table 3: **Ablation study on framework design**. We report Top-1 video retrieval results on UCF101 split 1. Here, $N$ is #training samples and $N_p$ is #fixation locations considered. See §4.3 for details.

| No. | Saccade | Pred. Err. | Reorg. | Mem. Size | Top-1 |
|-----|---------|-----------|--------|-----------|-------|
| 1   |         |           |        | $N$       | 39.4  |
| 2   |         |           |        | $N * N_p$ | 42.5  |
| 3   | ✓       |           |        | $N * N_p$ | 44.2  |
| 4   | ✓       | ✓         |        | $N * N_p$ | 45.0  |
| 5   | ✓       |           | ✓      | $N * N_p$ | 46.5  |
| 6   | ✓       | ✓         | ✓      | $N * N_p$ | **48.5** |

Table 4: **Ablation study on reorganization parameters** assessed by Top-1 retrieval accuracy on UCF101 split 1. Here, $R$ is #cluster results, and $Q^{(r)}$ is #clusters in the $r$-th result. * denotes updating prototypes every epoch. See §4.3.

| No. | $R$ | $Q^{(r)}$ | Top-1 | No. | $R$ | $Q^{(r)}$ | Top-1 |
|-----|-----|-----------|-------|-----|-----|-----------|-------|
| 1   | 0   |           | 40.8  | 7   | 3   | 500 1000 1500 | 41.7 |
| 2   | 1   | 500       | 41.5  | 8   | 3   | 1000 1000 1000 | 42.2 |
| 3   | 1   | 1000      | 42.1  | 9   | 3   | 1500 1500 1500 | **42.3** |
| 4   | 1   | 1500      | 42.0  | 10  | 3   | 1000 1500 2000 | 41.8 |
| 5   | 1   | 2000      | 41.9  | 11  | 3   | 1500 1500 1500 | 41.0* |
| 6   | 1   | 2500      | 41.3  | 12  | 3   | 1500 1500 1500 | 41.4* |

smaller than video SSL datasets such as HMDB51 and UCF101. Besides, the distribution of the videos in the saliency prediction datasets are typically not the same as that of the videos used for SSL training. Thus, it is hard for our SSL method trained on video saliency prediction datasets to achieve competitive performance when evaluated on downstream tasks.

To mitigate the above-mentioned problems, we resort to saliency models pre-trained on real gaze data for egocentric (Huang et al., 2018) or third-person videos (Droste et al., 2020), which are promising to provide an approximation of real gaze data. Considering that UCF101 contains third-person videos, we utilize UNISAL (Droste et al., 2020) to predict visual saliency maps for UCF101, and then use these maps to guide the construction of saccades during training. The fixation mask of a video clip is determined by the majority of the corresponding visual saliency maps. Since more than $90\%$ of the resulted fixation masks are center masks, to mitigate such distribution center bias, we randomly perturb the fixation mask labels with a probability of $0.5$. To assess the effectiveness of the saccades, we train the baseline with saccades for 300 epochs on UCF101. The model achieves $45.1\%$ Top-1 retrieval accuracy on UCF101, which is slightly better than $44.2\%$, the Top-1 accuracy of the one trained with artificial saccades as reported in Table 3. It is promising that a comparable amount of real gaze data would also bring in such benefits in our video SSL framework.

**Reorganization parameters.** To better reorganize the learned finer-grained semantics, we experiment on two key parameters introduced in §3.3, namely the number of cluster results $R$, the number of clusters in each result $Q^{(r)}$, as well as the clustering frequency and the number of warmup epochs. All the baselines are trained on UCF101 for 100 epochs using SGD. The learning rate is $0.1$ and is decayed by 5 at epoch $30, 60$ and $80$. The results are shown in Table 4. For the first ten baselines, the reorganization loss $\mathcal{L}_{\text{reorg}}$ is incorporated at epoch $61$, and the prototypes are updated every 5 epochs. For baseline 11 and 12, $\mathcal{L}_{\text{reorg}}$ is introduces at epoch $2$ and $61$, respectively, and the prototypes are updated every epoch. As can be observed, reorganization can consistently improve the Top-1 retrieval accuracy for a wide range of $R$ and $Q^{(r)}$. However, it is recommended to start prototypical learning later when the finer-grained semantics are relatively better captured, with less frequent prototype updates. In our full experiments where the models are trained for 300 epochs, we set $R = 3$ and $Q^{(r)} = 1500$ for $r = 1, \cdots, R$, and update prototypes every 5 epochs since epoch $181$.

## 5 CONCLUSION

In this work, we propose a video SSL method by taking inspiration from cognitive science and neuroscience on human visual perception. Instead of designing pretext tasks based on the inherent properties of videos, we explore the presence of saccades as an indicator of semantic change in a contrastive learning framework to mimic the role of self-awareness in human perception. To achieve semantic consistency in the absence of a saccade, we minimize the prediction error when using the state of a time point to predict that of another time point during a fixation. Finally, we strengthen the associations among similar representations through a post-learning reorganization process. Compared to previous contrastive learning based video SSL methods, our method learns more powerful representations by first making finer-grained distinctions for semantics in a video instance, and then associating similar semantics across different video instances through a reorganization process. Semantic consistency between the states of two time points within the same fixation is encouraged by minimizing the prediction error of using the earlier state to predict the later state. The proposed bio-inspired video SSL method achieves superior Top-1 video retrieval accuracy on UCF101 and outperforms other methods on the action recognition tasks.

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

## A APPENDIX

In this section, we present more details of saccades preparation in §A.1, and study more configurations on fixation masks in §A.2.

### A.1 DETAILS OF PREPARING SACCADES

The presence of a saccade indicates a semantic change in the stimuli that fall in the receptive field of the fovea (denoted as "fixation area" in our paper). In view of this, we proposed to manually determine such receptive fields using fixation masks. Thus, the change of the fixation mask, *i.e.*, a "saccade" in our design, reasonably indicates a semantic change of the stimuli in the fixation area, which inspires the design of our semantic-change-aware contrastive learning framework. The procedure for generating artificial saccades is detailed as follows.

We first determine the fixation locations by considering two aspects: i) the fixation locations are better to be evenly distributed since humans may attend to anywhere in the scene, and ii) the central region of the input shall be covered considering the center bias in free-viewing visual saliency (Tseng et al., 2009). To balance the performance and efficiency, in our experiments, we divide the mage into $2! \times 2$ grids, and take their centroids and the centroid of the entire image, which gives $N_p = 5$ locations in total. We also experiment on 9 fixation masks which corresponds to 9 fixation locations centered in the $3 \times 3$ grids, which is shown in §A.2.

Then, since the human eye can be viewed as an optical imaging system, we consider the *generalized pupil function* of an ideal imaging system and design the fixation mask as a binary mask where the

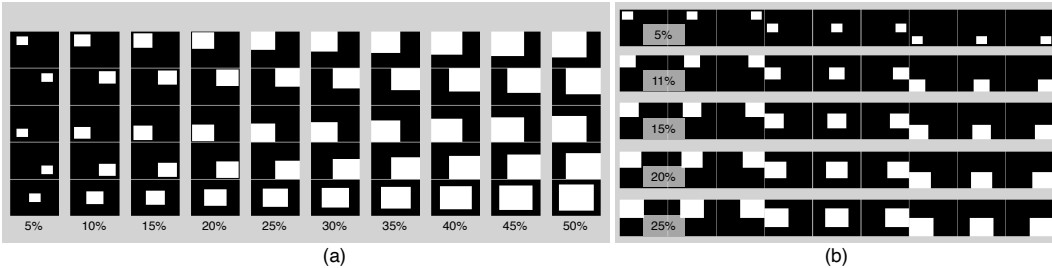

Figure 6: **Exemplar fixation masks** of various spatial sizes for (a) $N_p = 5$ and (b) $N_p = 9$, respectively. See §A.1 for more details.

Table 5: **Ablation study on fixation mask configurations.** We report Top-1 video retrieval results on UCF101 split 1. The best values are **boldfaced**. The value of default settings are underlined. See §A.2 for more discussions.

|  | Spatial size of fixation area | | | | | | | | | |
|---|---|---|---|---|---|---|---|---|---|---|
|  | 5% | 11% | 15% | 20% | 25% | 30% | 35% | 40% | 45% | 50% |
| $N_p = 5$ | 43.3 | 44.7 | 44.3 | 44.4 | 44.2 | 45.0 | **45.9** | 44.8 | 44.9 | 44.8 |
| $N_p = 9$ | 43.5 | 44.3 | 44.7 | **45.2** | 43.9 | | | | | |

value is 1 at every point within the fixation area and 0 otherwise. The fixation area, *i.e.*, the receptive field of the fovea, is represented using a rectangle centered at a fixation location that has the same aspect ratio as the input. In our experiments, we set the spatial size of the fixation area as 25% of the entire image size. We also explore the impact of different spatial sizes and numbers of fixations in §A.2. Exemplar fixation masks are shown in Fig. 6.

Finally, we construct an artificial saccade by manually assigning two different fixation masks to two video clips, which guarantees that the two fixation masks before and after an artificial saccade capture different visual receptive fields in the input scene. The two different fixation masks are randomly picked from $N_p$ pre-defined fixation masks.

## A.2 FURTHER STUDY ON FIXATION MASKS

In this section, we study the effect of the spatial size of the fixation areas and the number of fixations $N_p$ used for constructing artificial saccades. We train the baseline with saccades only instead of the full model on UCF101 split 1 for 100 epochs. The Top-1 retrieval accuracy on UCF101 of $N_p = 5$ and $N_p = 9$ with various spatial sizes are show in Table 5.

As can be observed, the Top-1 retrieval performance first increases and then decreases as the spatial sizes of the fixation area becomes larger. The optimal spatial sizes are 35% and 20% for $N_p = 5$ and $N_p = 9$, respectively. This is because that the information fall in the visual receptive field increases with the fixation area, and more information is beneficial for representation learning. However, when the overlap among different fixation areas enlarges, unintentional perturbations would be introduced, which impedes finer-grained semantic learning. Thus it is crucial to determine the optimal overlapping for different $N_p$ values.

