# OpenReview forum: "Bringing Saccades and Fixations into Self-supervised Video Representation Learning"
_ICLR.cc/2023/Conference — Submitted to ICLR 2023_

### Official Review · Reviewer_zkjU · 2022-10-23

**Confidence:** 2
**Correctness:** 3
**Technical Novelty And Significance:** 3
**Empirical Novelty And Significance:** 3
**Recommendation:** 6

**Clarity, Quality, Novelty And Reproducibility:**

While the paper is reasonably easy to read, the detail regarding the simulation of saccades is not explained well despite it being a key part of the proposed idea.

**Strength And Weaknesses:**

The proposed modules are relatively intuitive and as can be seen from the ablation studies (Tab. 3 and Tab. 4), the individual contributions seem to consistently improve performance as measured via top-1 retrieval acc. on UCF101.

Once it is understood, the idea of using “saccades” via masked inputs sounds reasonably sensible. However, the details of how it is done is not immediately apparent and the mentions of actual human saccades mis-leads the reader (as well as Fig. 1b). Further clarity in writing would be beneficial. No study of the no. of synthetic fixations is done - despite this procedure being introduced for the first time in this paper. The authors do not discuss whether real eye-tracking data could be used here, or whether pre-trained saliency models could be applied to the training process.

While the prototypical clustering method is reasonably well motivated, and Fig. 4 tells a meaningful story, one wonders why prototypical clustering was not built in as a more key component. That is, prototypes could have been updated every epoch, as is commonly done in methods such as DeepCluster and others.

**Summary Of The Paper:**

This paper proposes a scheme to improve self-supervised learning for videos. This is done by randomly generating synthetic “saccades” which are expressed as binary masks applied to the input images. This masking and a modified loss allows to better capture semantic changes. Furthermore, a loss for semantic consistency and a prototypical clustering method is proposed to further improve the method. The authors show consistent performance improvements for linear and fine-tuning evaluations of retrieval and action recognition, on the UCF101 and HMDB51 datasets.

**Summary Of The Review:**

The paper proposes contributions that can be understood fairly well, and backs up these proposals with strong experimental results. While the individual contributions may have been well-inspired, the written description on “saccades” and “prototypical contrastive learning” could be improved somewhat.

Edit post rebuttal: I appreciate the authors' efforts to respond to all concerns raised by the reviewers. My questions are partly addressed, and the paper's consistent performance improvements warrant a recommendation for acceptance. However, after carefully reviewing the other review texts and the authors' responses, I will not change my rating from 6. This is because the following changes would be better incorporated by a thorough re-write of the submission: (a) factors such as no. of fixations, spatial size of the fixation mask were analyzed only during the rebuttal phase, resulting in a different set of optimal parameters to the ones used in the paper, (b) the UniSAL approach show improvements over the arbitrarily designed "synthetic saccades" and probably should have been used as the main method of guiding positive/negative generation.

---

> ### Author Response · Authors · 2022-11-16
> **Response to Reviewer zkjU (Part 2 of 2)**
>
> **Q3. The authors do not discuss whether real eye-tracking data could be used here, or whether pre-trained saliency models could be applied to the training process.**
>
> A3. Thanks for pointing this out. We tried to incorporate real gaze data in our scheme by resorting to current video saliency prediction datasets such as DHF1K, Hollywood-2, and UCF sports. However,  those datasets typically contain no more than 1.5k training videos, which are much smaller than video SSL datasets such as HMDB51 and UCF101, containing 7k and 13k videos, respectively. Besides, the distribution of the videos in the saliency prediction datasets is typically not the same as that of the videos used for SSL training. Thus, it is hard for our SSL method trained on video saliency prediction datasets to achieve competitive performance when evaluated on downstream tasks.
>
> To mitigate the above-mentioned problems, we utilize a visual saliency prediction model [1*] pre-trained on real gaze data to predict visual saliency maps for UCF101, and then use these maps to guide the construction of saccades during training. The fixation mask of a video clip is determined by the majority of the corresponding visual saliency maps. Since more than 90% of the resulting fixation masks are center masks, to mitigate such distribution center bias, we randomly perturb the fixation mask labels with a probability of 0.5.  To assess the effectiveness of the saccades, we train the baseline with saccades only instead of the full model for 300 epochs on UCF101. The model achieves  45.1% Top-1 retrieval accuracy on UCF101, which is slightly better than 44.2%, the Top-1 accuracy of the baseline using the artificial saccades as reported in Table 3. It is promising that a comparable amount of real gaze data would also benefit our video self-supervised learning framework over artificial saccades.
>
> We have added the above discussions to Sec. 4.3 of the revised version.
>
> [1*] Droste et al, “Unified Image and Video Saliency Modeling”.  ECCV 2020
>
> **Q4. Why not update prototypes every epoch, as is commonly done in methods such as DeepCluster and others.**
>
> A4. We incorporate prototypical clustering as a post-learning process to redistribute the learned representations with finer-grained semantics. Our assumption is that incorporating prototypical clustering at the very beginning may introduce perturbations for learning finer-grained semantics. We choose to update the prototypes every five epochs to balance representation learning and training efficiency.
>
> Following the suggestions, we conducted two extra ablation experiments that start to update the prototypes every epoch from i) the 2nd epoch (following DeepCluster), to ii) the 61st epoch (following our previous setting). The models are trained for 100 epochs following the baselines in Table 4. We report the Top-1 retrieval accuracy on UCF101 below.
>
>
> | $R$ | $Q^{(r)}$ | Clust. Freq.| Start. Epoch|  Top-1  |
> |:---:| :-----: | :-----------: | :-----------: | :-------: |
> | 3 | 1500 1500 1500 |  5  |  61        | **42.3**|
> | 3 | 1500 1500 1500 |  1  |  2         | 41.0    |
> | 3 | 1500 1500 1500 |  1  |  61        | 41.4    |
>
>
> As can be observed, the two baselines updated every epoch achieve slightly inferior performance compared with our default settings. And later clustering outperforms earlier clustering. Thus, in our experiment, it is recommended to start prototypical learning later when the finer-grained semantics are relatively better captured, with less frequent prototype updates.
>
> We have added the above experiments and discussions to Sec. 4.3 in the revised version.
>
> **Q5. The written description on “saccades” and “prototypical contrastive learning” could be improved somewhat.**
>
> A5. We sincerely thank the reviewer for the insightful suggestions. We have improved the descriptions of “saccades” and “prototypical contrastive learning” in Sec. 3.1, Sec. 4.3 and Appendix A.1 in the revised version.

---

> ### Author Response · Authors · 2022-11-16
> **Response to Reviewer zkjU (Part 1 of 2)**
>
> **Q1. Once it is understood, the idea of using “saccades” via masked inputs sounds reasonably sensible. However, the details of how it is done is not immediately apparent and the mentions of actual human saccades mis-leads the reader (as well as Fig. 1b).**
>
> A2. Thanks for the insightful comments. The procedure for generating artificial saccades is detailed as follows. We first determine the fixation locations by considering the following two aspects: i) the fixation locations are better to be evenly distributed since humans may attend to anywhere in the scene, and ii) the central region of the input shall be covered considering the center bias in free-viewing visual saliency [1*]. To balance the performance and efficiency, in our experiments, we divide the image into 2x2 grids, and take their centroids and the centroid of the entire image, which gives 5 fixations in total. The five default fixation masks are shown in Fig. 3 (a).
>
> Then, since the human eye can be viewed as an optical imaging system, inspired by the generalized pupil function of an ideal imaging system, the **fixation mask** is designed as a binary mask where the value is 1 at every point within the fixation area and 0 otherwise. The fixation area, i.e., the receptive field of the fovea, is represented using a rectangle centered at a fixation location with the same aspect ratio as the input. In our experiments, we set the spatial size of the fixation area as 25% of the entire image size. We also explore the impact of different spatial sizes in Appendix A.2 of the revised version.
>
> Finally, we construct an **artificial saccade** by manually assigning two different fixation masks to two video clips, which guarantees that the two fixation masks before and after an artificial saccade capture different visual receptive fields in the input scene. The two different fixation masks are randomly picked from the five pre-defined fixation masks.  We have added the details to Appendix A.1 in the revised version.
>
> We would like to clarify that mentioning actual human saccades in Fig. 1b is more a conceptual illustration than a methodology presentation. We have modified the image caption of Fig. 1b to avoid such misunderstanding in the revised version.
>
> [1*] Tseng et al, “Quantifying center bias of observers in free viewing of dynamic natural scenes”, Journal of Vision 2009
>
> **Q2. No study of the no. of synthetic fixations is done - despite this procedure being introduced for the first time in this paper.**
>
> A2. Besides using 5 fixation masks, we also experiment on 9 fixation masks corresponding to 9 fixation locations centered in the 3x3 grids. The default spatial size of the fixation area is 1/9 of the entire input. We consider the baseline with saccades only instead of the full model to speed up the process. The Top-1 retrieval accuracy on UCF101 of 5 masks and 9 masks with various spatial sizes are shown below:
>
>
> |       | 5%   | 11%  | 15%  | 20%      | 25%         | 30%  | 35%      | 40%  | 45% | 50% |
> |-------|------|------|------|----------|-------------|------|----------|------|-----|-----|
> | $N_p=5$ | 43.3 | 44.7 | 44.3 | 44.4     | 44.2 | 45.0 | **45.9** | 44.8 |44.9 | 44.8|
> | $N_p=9$ | 43.5 | 44.3 | 44.7 | **45.2** | 43.9        |      |          |      |     |     |
>
>
> As can be observed, the Top-1 retrieval performance first increases and then decreases as the spatial sizes of the fixation area becomes larger. The optimal spatial sizes are 35% and 20% for $N_p=5$ and $N_p=9$, respectively. This is because the information falling in the visual receptive field increases with the fixation area, and more information is beneficial for representation learning. However, when the overlap among different fixation areas enlarges, unintentional perturbations would be introduced, which impedes finer-grained semantic learning. Thus it is crucial to determine the optimal overlapping for different $N_p$ values.

---

> ### Author Response · Authors · 2022-11-28
> **Thank you for the post-rebuttal comment**
>
> We sincerely thank the reviewer for all the valuable suggestions to improve our work and the positive assessment.
>
> Currently, we are running experiments of different spatial sizes on our full model to see if consistent improvements could be observed for $N_p=5$. We prefer $N_p=5$ to $N_p=9$ because the former generally achieves better performance with less computational cost according to our preliminary experiments during rebuttal. And continuing to increase the number of fixations $N_p$ might only provide limited benefits. We will update the corresponding experiment results and discussions in the final version.
>
> For incorporating pre-trained saliency models in saccades construction, we are applying this strategy to train the full model. We will update the results in the final version. Since our bio-inspired intuition lies more in utilizing the strong correlation between saccades and semantic changes to design a contrastive learning framework for video SSL, than designing a completely bio-inspired method to generate fixations or saccades during training, we might prefer to treat this strategy as an *alternative* instead of the *main method* to guide positive/negative generation. We leave it to the readers' decisions whether to utilize pre-trained saliency models or not based on their application scenarios. Both strategies are promising to achieve competitive video SSL performance.

---

### Official Review · Reviewer_aMU3 · 2022-10-24

**Confidence:** 4
**Correctness:** 4
**Technical Novelty And Significance:** 3
**Empirical Novelty And Significance:** 4
**Recommendation:** 8

**Clarity, Quality, Novelty And Reproducibility:**

This paper is understandable and of good quality technically.

There is also enough novelty in this work.

I believe the paper does not contain enough details for reproduction. Especially, the generation of fixation maps and the simulation of saccades are not clear.

**Strength And Weaknesses:**

The strengths of this work are as follows:
1. The bio-inspired self-supervised learning scheme is interesting and reasonable. Constructing artificial saccades rather than collecting data by eye-trackers is a clever and effective approach. This scheme is also proved by the experimental results.

2. The three parts of the proposed method are designed with reasonable motivation. There are also corresponding experiments in the next section to show the effect of each part.

3. The experiments are well-designed and enough to show the effectiveness of the proposed method.

I list several weaknesses of this paper, if revised, could make this work better:
1. Conducting experiments on larger-scale datasets can strengthen this work.

2. There is no experiment nor explanation of the spatial size and temporal length of the fixation mask. This size may have a great impact on the quality of self-supervised learning.

3. It is also good to discuss the use of pre-trained saliency/gaze prediction models in this self-supervised learning scheme, rather than the artificially generated saccades. Based on my knowledge, [A] is a related work that provides a gaze prediction model on videos consisting of the modeling of attention transition, which can be used as a simulation saccades.

[A] Huang et al, "Predicting Gaze in Egocentric Video by Learning Task-dependent Attention Transition", ECCV 2018.

**Summary Of The Paper:**

Inspired by the human visual perception that saccades usually indicate semantic changes in the visual receptive field, this paper proposes a self-supervised learning method by artificially generating masks that represent saccades, and learning through Semantic-change-aware contrastive learning, semantic consistency, and representation reorganization. The proposed self-supervised learning method shows
good performance on both video retrieval and action recognition tasks.

**Summary Of The Review:**

I think this paper proposes a novel and effective self-supervised learning scheme for videos. The method is reasonable and experiments are well conducted. I don't see major weaknesses in this paper.

After the revision, I believe this work becomes clear and the experiments become more solid. Thus I am raising my recommendation to accept.

---

> ### Author Response · Authors · 2022-11-16
> **Response to Reviewer aMU3**
>
> **Q1. Conducting experiments on larger-scale datasets can strengthen this work.**
>
> A1. Thanks for the suggestion. Unfortunately, experiments on larger-scale datasets like Kinetics [1*] require much higher computational resources to complete within the rebuttal period than what our platform currently supports. We, therefore, leave it to future work. In our humble opinion, the experiments on two representative datasets (i.e., UCF101 and HMDB51) have shown the feasibility and potential of the proposed video SSL method.
>
> [1*] Carreira and Zisserman, “Quo vadis, action recognition? a new model and the kinetics dataset”.  CVPR 2017
>
> **Q2. There is no experiment nor explanation of the spatial size and temporal length of the fixation mask. This size may have a great impact on the quality of self-supervised learning.**
>
> A2. Thanks for pointing this out. In our experiments, the spatial size of the fixation area (values equal to 1) in the mask is 1/4 of that of the whole input, and the temporal length of the fixation mask is 16 to fit the clip length requirement of the feature embedding network. We have added more detailed explanations in Sec. 3.1 and Appendix A.1 of the revised version.
>
> We conduct a further study on different spatial sizes of the fixation area. We trained the baseline model with saccades only instead of the full model for 300 epochs to speed up the process. The experiment results are shown in the table below.
>
>
> |       | 5%   | 11%  | 15%  | 20%  | 25%         | 30%  | 35%      | 40%  | 45% | 50% |
> |-------|------|------|------|------|-------------|------|----------|------|-----|-----|
> | $N_p=5$ | 43.3 | 44.7 | 44.3 | 44.4 |44.2 | 45.0 | **45.9** | 44.8 |44.9 | 44.8|
>
>
> As can be observed, the Top-1 retrieval performance first increases and then decreases as the spatial sizes of the fixation area becomes larger. This is because the information fall in the visual receptive field increases with the fixation area, and more information is beneficial for representation learning. However, when the overlap among different fixation areas enlarges, unintentional perturbations would be introduced, which impedes finer-grained semantic learning. The optimal spatial size is 35% for five fixation masks instead of the default 25%, showing potential room for improvement. We leave this as future work. In the revised version, we have also added related experiments and discussions in Appendix A.2.
>
> **Q3. Discussion on the usage of pre-trained saliency/gaze prediction models.**
>
> Thanks for the insightful suggestion. We agree that saliency models pre-trained on real gaze data for egocentric (first-person) videos [A] or third-person videos [2*] are promising to provide an approximation of real gaze data. Considering that UCF101 contains third-person videos, we utilize [2*] to predict visual saliency maps for UCF101, and then use these maps to guide the construction of saccades during training. The fixation mask of a video clip is determined by the majority of the corresponding visual saliency maps. Since more than 90% of the resulting fixation masks are center masks, to mitigate such distribution center bias, we randomly perturb the fixation mask labels with a probability of 0.5. To assess the effectiveness of the saccades, we train the baseline with saccades only instead of the full model for 300 epochs on UCF101. The model achieves 45.1% Top-1 retrieval accuracy on UCF101, which is slightly better than 44.2%, the Top-1 accuracy of the baseline using the artificial saccades as reported in Table 3. It is promising that a comparable amount of real gaze data would also benefit our video self-supervised learning framework over artificial saccades. More discussions can be found in Sec. 4.3 of the revised version.
>
> [A] Huang et al, "Predicting Gaze in Egocentric Video by Learning Task-dependent Attention Transition", ECCV 2018.
>
> [2*] Droste et al, “Unified Image and Video Saliency Modeling”.  ECCV 2020
>
> **Q4. I believe the paper does not contain enough details for reproduction. Especially, the generation of fixation maps and the simulation of saccades are not clear.**
>
> A4. Thanks for pointing this out. We have added more details about the fixation maps and saccades in Sec. 3.1 and Appendix A.1 of the revised version, which we hope could ease the reproductivity of our method. We have also uploaded our test code and pre-trained models as supplementary material. We will release our full code in future.

---

> ### Author Response · Authors · 2022-11-28
> **Thank you for the post-rebuttal comment**
>
> We sincerely thank the reviewer for all the valuable suggestions to improve our work and the positive assessment.
> &nbsp;
>
> Best,
>
> Authors.

---

### Official Review · Reviewer_ucm9 · 2022-10-25

**Confidence:** 4
**Correctness:** 2
**Technical Novelty And Significance:** 3
**Empirical Novelty And Significance:** 3
**Recommendation:** 5

**Clarity, Quality, Novelty And Reproducibility:**

The idea is quite interesting and the proposed model has some novel ideas. However, the technical part is not very clearly presented.

**Strength And Weaknesses:**

Strengths:

--The idea of in-cooperating saccades/fixation into SSL video representation learning is very interesting.

--And the proposed model contain some novelty, e.g, contrastive learning with fixations.

--There are not only experiments for classification, but also retrieval, showing the advantages of the learned representations. And the experiments are conducted on two data sets, with comparison to multiple existing video SSL methods (including contrastive ones and non-contrastive ones), making it quite convincing.


Weakness

--Some technical part is not clear. For example, how are the 5 fixations created ? The authors only mention "simulate the pupil function of an ideal system" without any explanation in details.

--Further more, for two different frames in the same video, how do we judge whether two fixations are the same, e.g., when two fixation masks are at the same location (top right corner), or two fixation masks contain the same object, or some other way?

**Summary Of The Paper:**

This paper proposed a new self-supervised learning model for video representation by introducing the idea of saccades/fixations. More specifically, a semantic change aware contrastive learning is proposed such that a positive pair is from the same fixation region of the same video, followed by reorganization of prototypical contrastive learning. Further more,  semantic consistency learning module is also included to encourage the consistency between the states of two time points within the same fixation, by minimizing the prediction error of using earlier state to predict later state.

Experiments are conducted for both video retrieval and video action recognition, on UCF101 and HMDB51 data sets.

**Summary Of The Review:**

Overall, the proposed idea is interesting and novel to some extent. The experiment results are also encouraging. However, some key technical aspect is unclear, making it unconvincing, so I can not give high rating for now.

---

> ### Author Response · Authors · 2022-11-16
> **Response to Reviewer ucm9**
>
> **Q1. Some technical part is not clear. For example, how are the 5 fixations created ? The authors only mention "simulate the pupil function of an ideal system" without any explanation in details.**
>
> A1. Thanks for pointing this out. We determine the fixation locations by considering the following two aspects: i) the fixation locations are better to be evenly distributed since humans may attend to anywhere in the scene, and ii) the central region of the input shall be covered considering the center bias in free-viewing visual saliency [1*]. To balance the performance and efficiency, in our experiments, we divide the image into 2x2 grids, then take their centroids and the centroid of the entire image, giving a total of 5 fixations.
>
> Since the human eye can be viewed as an optical imaging system, inspired by the generalized pupil function of an ideal imaging system, the fixation mask is designed as a binary mask where the value is 1 at every point within the fixation area and 0 otherwise. The fixation area, i.e., the receptive field of the fovea, is represented using a rectangle centered at a fixation location with the same aspect ratio as the input. In our experiments, we set the spatial size of the fixation area as 25% of the entire image size. We have also added the technical details to the revised version in Appendix A.2.
>
> [1*] Tseng et al, “Quantifying center bias of observers in free viewing of dynamic natural scenes”, Journal of Vision 2009
>
> **Q2. Further more, for two different frames in the same video, how do we judge whether two fixations are the same, e.g., when two fixation masks are at the same location (top right corner), or two fixation masks contain the same object, or some other way?**
>
> A2. We do not need to judge whether two fixations of two different frames in the same video are the same during training. This is because we construct an artificial saccade by manually assigning two different fixation masks to two different frames (or, more accurately, two different clips) in the same video. This guarantees that the two fixation masks before and after an artificial saccade capture different visual receptive fields in the input scene, i.e., there is a change of the fixations before and after an artificial saccade. The two different fixation masks are randomly picked from the five pre-defined fixation masks.

---

### Official Review · Reviewer_MEj8 · 2022-10-25

**Confidence:** 2
**Correctness:** 3
**Technical Novelty And Significance:** 2
**Empirical Novelty And Significance:** 2
**Recommendation:** 5

**Clarity, Quality, Novelty And Reproducibility:**

I find the paper not clear and not easy to read and follow, given lots of vague sentences and definitions, I find it hard to reproduce.
As I understand the main novelty of paper is to use artificial fixation and saccades to generate negative and positive samples, which I find then the novelty being very marginal.







**Strength And Weaknesses:**

The idea of using gaze data to improve the SSL is interesting and previously also has been shown promising results. The paper covered related work in depth and gave good explanation upon each work.

However I have a hard time reading and following the paper, I find that the paper has some vague sentences or references to some papers with not clear justifications. Fro example on page 4 "To
this end, we treat such representations as negative pairs and otherwise positive pairs and optimize
the latent feature space by minimizing the contrastive loss." I really have a hard time understanding such vague sentences..

I am not sure why presence of saccade should indicates a semantic change in the video, in real human gaze data we performs saccades to cover different parts of stimuli and it does not necessarily means a semantic change in the stimuli or the scene, as simply human eye fovea can not cover the whole scene in details as once.  Hence comparing the work to a biological inspired approach is over reaching as it has not been tested using real gaze data.

I could not find the details how the fixation locations was generated, why 5 fixations were selected or what is the size of the mask, are these locations generated randomly? or a saliency based approach is used to generate fixations data?

How is the v_ik ' generated? What does it mean it is an augmented version of vik?

Overall, I was not able to understand the paper fully as it was not written clearly and I could not really make sense of the results.
It is also not clear if the overall performance is due to the using more informative labels to generate positive and negative pairs?


**Summary Of The Paper:**

The paper proposed a self-supervised video representation learning that would take advantage of artificial cascades and fixation points mimicking human eye behavior.  Using the generated fixation and saccades, they could generate  positive and negative samples for each video frame that is used later in a contrastive learning approach to bring similar examples closer and negative ones further away.



**Summary Of The Review:**

I think using human gaze data could be a potential use case in SSL approaches, and the idea of the paper is interesting and promising for future research. However, this paper is not written clearly, and I personally find it very hard to read and follow.

---

> ### Author Response · Authors · 2022-11-16
> **Response to Reviewer MEj8 (Part 2 of 2)**
>
> **Q3. I could not find the details how the fixation locations was generated, why 5 fixations were selected or what is the size of the mask, are these locations generated randomly? or a saliency based approach is used to generate fixations data?**
>
> A3. We determine the fixation locations by considering the following two aspects: i) the fixation locations are better to be evenly distributed since humans may attend to anywhere in the scene, and ii) the central region of the input shall be covered considering the center bias in free-viewing visual saliency [4*]. To balance the performance and efficiency, in our experiments, we divide the image into 2x2 grids, and take their centroids and the centroid of the entire image as the set of fixation locations, which gives 5 fixations in total.
>
> Since the human eye can be viewed as an optical imaging system inspired by the generalized pupil function of an ideal imaging system, the fixation mask is designed as a binary mask where the value is 1 at every point within the fixation area and 0 otherwise. The fixation area, i.e., the receptive field of the fovea, is represented using a rectangle centered at a fixation location with the same aspect ratio as the input. In our experiments, we set the spatial size of the fixation area as 25% of the entire image size. We also explore the impact of different spatial sizes in Appendix A.2 of the revised version.
>
> Please note that we do not use other saliency-based approaches to generate fixation data in our initial version. To study this problem, in our revised version, we utilize pre-trained saliency models to generate fixation data training and achieve slightly better results than using artificial saccades. More details are shown in Sec. 4.3.
>
> [4*] Tseng et al, “Quantifying center bias of observers in free viewing of dynamic natural scenes”, Journal of Vision, 2009
>
> **Q4. How is the v_ik ' generated? What does it mean it is an augmented version of v_ik?**
>
> A4. We first obtain x_i’ by applying commonly-used data augmentations on x_i, and then obtain v_ik’ by applying the embedding function f_{\theta} on x_i’ masked by the fixation mask m_k. We have made this point clearer in the revised version.
>
> **Q5. It is also not clear if the overall performance is due to the using more informative labels to generate positive and negative pairs?**
>
> A5. Please note that we have **NOT** used any additional informative labels during training, and the positive and negative pairs are generated using artificial saccades. We attribute our improvements over previous contrastive-based video SSL methods to making a finer-grained distinction of the semantics from the same video instance. Semantic consistency learning and representation reorganization also contribute to improving the learned representations.
>
> **Q6. I find the paper not clear and not easy to read and follow, given lots of vague sentences and definitions, I find it hard to reproduce. As I understand the main novelty of paper is to use artificial fixation and saccades to generate negative and positive samples, which I find then the novelty being very marginal.**
>
> A6. We have improved many statements as suggested to facilitate the understanding of our method. To ensure reproducibility, we have uploaded our test code and pre-trained models as supplementary material. We will release our full code in future.
>
> We would like to emphasize that the novelty of our paper is to propose a video SSL scheme by taking inspiration from cognitive science and neuroscience on human visual perception, especially on the strong correlation between saccades and semantic changes during watching. Our method consists of semantic-change-aware learning, semantic consistency learning, and representation reorganization. Compared to previous deep video SSL methods that typically learn discriminative representations by considering the inherent data properties, our method makes distinctions within the same video indicated by saccades and achieves finer-grained semantic learning with consistency. The association among the learned finer-grained semantics is further enhanced to facilitate grouping unseen input stimuli into meaningful categories based on similarity.

---

> ### Author Response · Authors · 2022-11-16
> **Response to Reviewer MEj8 (Part 1 of 2)**
>
> **Q1. Vague sentences or references like “To this end, we treat such representations as negative pairs and otherwise positive pairs and optimize the latent feature space by minimizing the contrastive loss.”**
>
> A1. Thanks for pointing this out. We have gone through the paper to improve the clarity. E.g., we rewrite the above-mentioned sentence as “To this end, we treat the representations before and after a saccade as negative pairs, and otherwise positive pairs. The latent feature space is thus optimized by minimizing the contrastive loss”. More improvements can be found in the revised version.
>
> **Q2. I am not sure why presence of saccade should indicates a semantic change in the video, in real human gaze data we performs saccades to cover different parts of stimuli and it does not necessarily means a semantic change in the stimuli or the scene, as simply human eye fovea can not cover the whole scene in details as once. Hence comparing the work to a biological inspired approach is over reaching as it has not been tested using real gaze data.**
>
> A2. Thanks for the insightful comments. We agree with the reviewer that a saccade does not necessarily mean a semantic change in the stimuli or the scene, e.g., for close-up views. However, a strong correlation exists between saccades and semantic changes, especially when there are multiple objects in the scene [1*]. This is because the change of foveate positions (a saccade) is modulated by top-down visual attention in the brain, and the semantic change in the scene would affect the attention deployment, which drives saccades [2*].
>
> We tried to incorporate real gaze data in our scheme by resorting to current video saliency prediction datasets such as DHF1K, Hollywood-2, and UCF sports. However,  those datasets typically contain no more than 1.5k training videos, which are much smaller than video SSL datasets such as HMDB51 and UCF101, containing 7k and 13k videos, respectively. Besides, the distribution of the videos in the saliency prediction datasets is typically not the same as that of the videos used for SSL training. Thus, it is hard for our SSL method trained on video saliency prediction datasets to achieve competitive performance when evaluated on downstream tasks.
>
> To mitigate the above-mentioned problems, we utilize a visual saliency prediction model [3*] pre-trained on real gaze data to predict visual saliency maps for UCF101. We then use these maps to guide the construction of saccades during training. The fixation mask of a video clip is determined by the majority of the corresponding visual saliency maps. Since more than 90% of the resulting fixation masks are center masks, to mitigate such distribution center bias, we randomly perturb the fixation mask labels with a probability of 0.5.  To assess the effectiveness of the saccades, we train the baseline with saccades only instead of the full model for 300 epochs on UCF101. The model achieves 45.1% Top-1 retrieval accuracy on UCF101, which is slightly better than 44.2%, the Top-1 accuracy of the baseline using the artificial saccades as reported in Table 3. It is promising that a comparable amount of real gaze data would also benefit our video self-supervised learning framework over artificial saccades. More discussions can be found in Sec. 4.3 of the revised version.
>
> [1*] Weaver et al, “The effect of semantic information on saccade trajectory deviations”, Vision Research, 2011
>
> [2*] Wu et al, “Guidance of visual attention by semantic information in real-world scenes”, Frontiers in Psychology, 2014
>
> [3*] Droste et al, “Unified image and video saliency modeling”,  ECCV 2020

---

> > ### Comment · Reviewer_MEj8 · 2022-11-27
> > **I am not convinced that they method to generate fixation is human gaze inspired and still the paper need improvement in writing.**
> >
> > there are different approaches for saliency and fixation location prediction that try to overcome central biases, so rather than randomly jittering the fixation mask I would try out different saliency map prediction approaches first.
> >
> > going through the paper again, I still find the paper ambiguous, not well written. I have hard time to really associate their method to gaze inspired approach. Hence I keep my rating the same as before.

---

> > > ### Author Response · Authors · 2022-11-28
> > > **Thank you for the post-rebuttal comment**
> > >
> > > We sincerely thank the reviewer for all the valuable suggestions to improve our work.
> > >
> > > The bio-inspired intuition lies more in utilizing the strong correlation between saccades and semantic changes to design a contrastive learning framework for video SSL, than designing a completely bio-inspired method to generate fixations or saccades during training. For constructing artificial saccades, our approach shares the insight with another bio-inspired SSL method, CLAPP [1*] (as discussed in Sec. 2 in our paper), wherein the saccade is constructed by alternating the video instances during training. The experiment results proved the effectiveness of our strategy.
> > >
> > > We appreciate the suggestions of incorporating the pre-trained saliency models during training, and the preliminary results are promising. To make our method more complete, we are working on applying this new strategy of generating fixations to our full model. We will incorporate results and discussions on the two alternative ways of constructing saccades in the final version. In this way, the readers could choose to utilize either the pre-trained saliency models or the designed fixations according to their application scenarios and requirements.
> > > &nbsp;
> > >
> > > [1*] Illing et al, “Local plasticity rules can learn deep representations using self-supervised contrastive predictions”, NeurIPS 2021

---

### Author Response · Authors · 2022-11-16
**General Response**

We sincerely thank the reviewers for their valuable and insightful feedback.  We are pleased that the reviewers found our idea interesting (MEj8, ucm9, aMU3), promising (MEj8) and novel (ucm9, aMU3), module design intuitive (zkjU),  and experimental results encouraging (ucm9, aMU3, zkjU).

In the revised version, we have made the following improvements:

- We improved the clarity of some sentences and definitions as suggested.
- We explained the relation between saccades and semantic changes in the video.
- We added details about constructing fixation masks and saccades and clarified the technical part.
- We added experiments and discussions on utilizing real gaze data or a pre-trained saliency model for training.
- We added a further study on the spatial sizes of the fixation area and the number of fixation locations used for constructing artificial saccades.

Due to space limitations, not all comments could be addressed in the main paper and had to be deferred to the appendix. Changes in the revised version are marked in blue.

We appreciate all the suggestions made by reviewers to improve our work. We are looking forward to further feedback.

---

### Decision · Program_Chairs · 2023-01-20

**Decision:**

Reject

**Justification For Why Not Higher Score:**

Major:
- Lacking a thorough empirical evaluation, both in terms of datasets and missing ablations (spatial size and temporal length of the fixation mask), comparison to pre-trained saliency/gaze prediction models,
- Clarity and presentation of the technical aspects of the work.
- Novelty, if the method is primarily about a new way to generate positive and negative pairs -- this depends on the thorough empirical evaluation.

Minor:
- Unclear, or at least, non-verifiable intuition that saccades actually imply semantic changes.


**Justification For Why Not Lower Score:**

N/A

**Metareview: Summary, Strengths And Weaknesses:**

The authors propose a new self-supervised learning model for video representation learning based on the idea of saccades with the goal of better capturing semantic changes across frames ("saccades usually indicate semantic changes in the visual receptive field", https://en.wikipedia.org/wiki/Saccade). To implement this idea, the authors express synthetic saccades as binary masks applied to the input images and apply prototypical contrastive learning coupled with a semantic consistency module. The authors evaluate the proposed method on several video retrieval and action recognition tasks where it achieves competitive performance both for fine-tuning and linear probes.

The reviewers appreciated the idea of using gaze data to inform a SSL method and the placement of this paper relative to existing work. Reviewers found that the submission was lacking: (1) A thorough empirical evaluation, both in terms of datasets and missing ablations (spatial size and temporal length of the fixation mask), comparison to pre-trained saliency/gaze prediction models, (2) Clarity and presentation of the technical aspects of the work. (3) Unclear, or at least, non-verifiable intuition that saccades actually imply semantic changes. (4) Novelty, if the method is primarily about a new way to generate positive and negative pairs -- this depends on the thorough empirical evaluation in (1).

This is a borderline submission. Taking into account the rebuttal and the discussion, I feel that the empirical evaluation needs to be more through and a major revision is necessary for acceptance to ICLR. The authors should focus on resolving (1) given that it's the main contribution of the paper and the novelty (beyond the connection to biological mechanisms) is limited.